# A Hybrid Method for Keystroke Biometric User Identification

**Md L. Ali** [1,*] **, Kutub Thakur** [2] **and Muath A. Obaidat** [3]

[1] Department of Computer Science & Physics, Rider University, 2083 Lawrenceville Rd, Lawrenceville, NJ 08648, USA
[2] Department of Professional Security Studies, New Jersey City University, Jersey City, NJ 07305, USA
[3] Department of Computer Science, City University of New York, New York, NY 10019, USA
* Correspondence: mdali@rider.edu

**Abstract:** The generative model and discriminative model are the two categories of statistical models used in keystroke biometric areas. Generative models have the trait of handling missing or irregular data, and perform well for limited training data. Discriminative models are fast in making predictions for new data, resulting in faster classification of new data compared to the generative models. In an attempt to build an efficient model for keystroke biometric user identification, this study proposes a hybrid POHMM/SVM method taking advantage of both generative and discriminative models. The partially observable hidden Markov model (POHMM) is an extension of the hidden Markov model (HMM), which has shown promising performance in user verification and handling missing or infrequent data. On the other hand, the support vector machine (SVM) has been a widely used discriminative model in keystroke biometric systems for the last decade and achieved a higher accuracy rate for large data sets. In the proposed model, features are extracted using the POHMM model, and a one-class support vector machine is used as the anomaly detector. For user identification, the study examines POHMM parameters using five different discriminative classifiers: support vector machines, k-nearest neighbor, random forest, multilayer perceptron (MLP) neural network, and logistic regression. The best accuracy of 91.3% (mean 0.868, SD 0.132) is achieved by the proposed hybrid POHMM/SVM approach among all generative and discriminative models.

**Keywords:** generative model; discriminative model; keystroke biometrics; identification; hybrid model

## 1. Introduction

In this modern age, all businesses, small or large, organizations, and even governments are heavily relying on computerized systems to process and store their important and sensitive information. However, attackers are all around and trying to gain unauthorized access to individuals' information as well as companies' databases. This makes cybersecurity a primary goal to safeguard sensitive information from any unauthorized access. User authentication is the most important and crucial process to verify a person's or system's identity during the access control step. There are three traditional modes used for authentication of a person: possessions, knowledge, and biometrics. Possessions include what we have, such as keys, passports, smartcards, etc., which can be shared, can be duplicated, or even may be lost or stolen. Knowledge includes what we know such as secret information (i.e., passwords) and is most widely used. However, many passwords are easy to guess, might be shared, or may be forgotten. On the other hand, biometrics includes what we are or what we do, i.e., metrics related to human characteristics and traits. Biometric identifiers can be classified as physiological and behavioral characteristics. Biometric identifiers are not possible to share, difficult to reproduce, and cannot be lost or stolen [1]. Password-based authentication is the most common method used to protect data from unauthorized users. Many people choose short and weak passwords, such as birth date, social security number, some dictionary word, etc. Attackers can easily crack

weak passwords using brute-force or dictionary attacks. Password best practices suggest that users use strong and complex passwords that contain combinations of uppercase letters, lowercase letters, special characters, and digits. Although complex passwords are difficult to crack for the attackers, they are also difficult for the users to remember. Therefore, additional mechanisms are needed to enhance the security of password-based authentication. Keystroke biometrics is a type of behavioral biometrics that can add an extra layer of security with password-based applications. It can also perform continuous authentication of a user, such as continuously monitoring typing behavior of a user during online examination [2]. The accomplishment of a keystroke biometric system is ascertained by how well the system can distinguish authentic users from the impersonators. There are two main functions of biometric systems—identification and verification. Identification, also known as user recognition, is where the system classifies an unknown user from N known users (1:N match). Verification, also known as authentication, is where the system validates or proves the identity provided by the user during the identification process (1:1 match). Authentication can be static and/or a continuous authentication. Static authentication can be used with traditional username- and password-based authentication. Short and fixed text is used in static authentication, while long and free text is used in continuous authentication. This research focuses on keystroke biometric user identification and uses short and fixed text.

Over the past few years, several machine-learning models have been used for biometric applications. Some models have been proven to have good fit and excellent performance with specific applications. The main goal of supervised machine learning is to develop models that have the ability to learn and generalize from previously observed data, e.g., $(a_1,b_1), \ldots (a_n,b_n)$, where $a$ is the observation sequence and $b$ is the corresponding class, allowing predictions based on evidence in the presence of uncertainty. Statistical models for classification broadly fall into two categories: the generative and discriminative models. The generative models approximate the joint distribution, $p(a,b)$, over the observation sequences and make classification decisions by computing posterior probabilities based on Bayes's rule, $P(b|a) = \frac{p(a|b)P(b)}{p(a)}$ where $p(b)$ is the prior probability and $p(a \mid b)$ is the likelihood probability [3]. Popular generative models are Gaussians, naïve Bayes, mixture of multinomial, mixture of Gaussians, hidden Markov model (HMM), Markov random fields, probabilistic context-free grammar, averaged one-dependence estimators, latent dirichlet allocation (LDA), restricted Boltzmann machine [4], generative adversarial network, and sigmoidal belief networks.

Unlike generative models, discriminative models directly model the mapping from observation sequences to label sequences as a posterior distribution $P(b \mid a)$ or as a discriminant function $f_b(a)$. Therefore, discriminative models do not need to maintain valid likelihood or prior distributions [5]. For instance, logistic regression machines and conditional random fields (CRF) directly model the posterior of the label sequence from the given observations, while support vector machines directly model the discriminative function or decision boundaries between classes [6]. Common discriminative models are logistic regression, support vector machines, boosting, conditional random fields, linear regression, neural networks, and random forest.

The hidden Markov model (HMM) and its extensions have been applied in speech recognition, natural language processing, human activity recognition, handwriting recognition, shape recognition, face and gesture recognition, and many others. Although HMM outperforms other models in speech, signature, and gesture recognition, its performance is poor in keystroke biometric (KB) systems. Moreover, there seems to be limited research [7–12] conducted using HMMs in KB systems. Monaco and Tappert [13] proposed an extension of the hidden Markov model called the partially observable hidden Markov model (POHMM), which shows significant improvement in performance when dealing with different data sets. Support vector machines (SVMs) are one of the most popular discriminative anomaly detectors of contemporary time and have been successfully applied to a variety of physiological and behavioral biometrics. SVMs are considered excellent

anomaly detectors with proficient speed. Various studies suggest that for the large data sets in keystroke biometric systems, SVMs perform better than the neural network in terms of accuracy and computational complexities, but exhibit poor performance in handling missing or infrequent data. This research reviews different generative and discriminative models used in KB systems and proposes a hybrid generative/discriminative model to take advantage of both generative and discriminative models and to increase overall model performance. In the experimental part of this work, the performance of a proposed generative/discriminative model has been evaluated using publicly available CMU [14] short fixed-text keystroke data sets. The identification accuracy of the proposed model has been compared with different generative and discriminative models. The contributions of this research are presented below.

1. Keystroke dynamics has certain advantages over other biometrics. However, the main disadvantage of keystroke dynamics is lower accuracy. By understanding the performance and limitations of different existing systems, this study proposes a model to obtain improved accuracy and performance in biometric systems.
2. This research evaluates the generative models POHMM and HMM and a discriminative model, SVM, along with the proposed model with keystroke short fixed-text data set and follows the same evaluation procedure for objective comparison with other novelty detectors.
3. For user identification, this study examined different POHMM parameters using five different classifiers: support vector machines, k-nearest neighbors, random forest, multilayer perceptron (MLP) neural network, and logistic regression, and compared identification accuracy with other existing models.

This paper is organized as follows. Section 2 highlights the related work and contribution of this paper. Section 3 presents the proposed model descriptions, experimental setup, and evaluation procedures. Section 4 discusses the experimental results and comparison with other models to validate the proposed model. Section 5 summarizes the findings of the experiment, and finally, Section 6 contains the conclusion and some possible directions for future work.

## 2. Related Work

All applications of machine learning focus on one of two goals: prediction or interpretation. There is a tradeoff between accurate prediction and interpretation. Generative and discriminative models play a huge role in illuminating the tradeoff between interpretability and performance. Generative models build a full model of the distribution of the features for each of the two classes and then compare the differences. On the other hand, the discriminative approach focuses on correctly modeling just the boundary between two classes. Discriminative approaches follow modeling the conditional distribution of the label on the input features, where generative approaches follow joint distribution of the labels and features [2,3,13,14].

Generative models have several advantages over discriminative models. A generative model of the data implies that model parameters have well-defined semantics in relation to the generative process. Moreover, generative models are frequently stated in terms of a probabilistic framework, which helps to handle missing data [15]. The most important benefit in discriminative models is better performance with large or infinite training data [3]. For a small amount of training data, generative models usually perform better than the discriminative models. However, for a large amount of labeled training data, discriminative models outperform generative models.

Although keystroke biometrics is relatively new in the study of biometrics, there are many research papers related to this topic. It is a very daunting task to compare most of these studies, because different experiments have used different data sets, and some have used their own unique data set [6,7,9,16,17]. Again, they also differ in terms of features, classifiers, and evaluation methods. Most of the research conducted on keystroke biometrics has focused on user verification (authentication) compared to user identification. This study

has made a performance comparison of several popular generative and discriminative models used in KB systems. Research on those that have achieved better performance than others in the same area have been included in the comparison tables. Table 1 compares research that achieved improved performance using different discriminative models in keystroke biometrics for anomaly detection. Table 2 compares research that sought to show promising performance using generative models in KB systems.

**Table 1.** Classification based on popular discriminative models.

| Classifiers and References | Number of Subjects | Results |
| --- | --- | --- |
| Random Forest [18] | 10 | FAR: 0.41, FRR: 0.63, EER: 0.53 |
| Random Forests [19] | 41 | EER: 2.00 |
| Random Forest [20] | 21 | FAR: 3.47, FRR: 0.00, EER: 1.73 |
| Random Forest [21] | 28 | FAR: 0.03, FRR: 1.51, EER: 1.00 |
| Neural Network [22] | 100 | FAR: 1.00, FRR: 8.00 |
| Neural Network [23] | 10 | EER: 1.00 |
| Neural Network [24] | 20 | FAR: 4.12, FRR: 5.55 |
| Neural Network [25] | 22 | FAR: 2.00 |
| Neural Network [26] | 24 | FAR: 8.00, FRR: 9.00 |
| Neural Network [27] | 30 | FAR: 2.00 |
| Neural Network [28] | 151 | FAR: 1.10, FRR: 0.00 |
| Support Vector Machine [29] | 24 | EER: 2.00 |
| Support Vector Machine [30] | 24 | Far: 0.76, FRR: 0.81, EER: 1.57 |
| Support Vector Machine [31] | 39 | EER: 0–2.94 |
| Support Vector Machine [32] | 25 | EER: 1.00 |
| *k*-Nearest Neighbors [33] | 63 | Accuracy: 83.22–92.14 |
| *k*-Nearest Neighbors [34] | 30 | EER: 0.50 |
| Convolution neural network (CNN) [35] | 148 | Accuracy: 97.75 |
| Random Forest [36] | - | Accuracy: 86 |
| CNN [37] | 148 | Accuracy: 82 |

FAR: False Acceptance Rate, FRR: False Rejection Rate, EER: Equal Error Rate.

**Table 2.** Classification based on popular generative models.

| Classifiers and References | Number of Subjects | Results |
| --- | --- | --- |
| Naïve Bayesian [38] | 26 | FAR: 2.80, FRR: 8.10 |
| Naïve Bayesian [39] | 33 | EER: 1.72 |
| Naïve Bayesian [40] | 16 | EER: 4.28 |
| Gaussian Mixture Model [41] | 41 | FAR: 4.3, FRR: 4.8, EER: 4.4 |
| Gaussian Mixture Model [42] | 10 | FAR: 2.10, FRR: 2.40 |
| Hidden Markov Model [10] | 20 | EER: 3.60 |
| Hidden Markov Model [9] | 58 | EER: 2.54 |
| Weighted mean, Standard deviation [33] | 31 | Accuracy: 90 |
| Deep Neural Network [43] | 51 | EER: 0.35 |
| Deep Learning [44] | 51 | Accuracy: 92.60 |

FAR: False Acceptance Rate, FRR: False Rejection Rate, EER: Equal Error Rate.

Most of the research has followed the static authentication mode, which attempts to verify users at the initial interaction with the system. Only a few studies have focused on dynamic authentication where the system continuously or periodically monitors the keystroke behavior. A majority of the experiments had no input freedom, which means all users shared the same password [2,11–15]. Researchers have used discriminative models mainly for large samples, and for smaller samples, generative models were preferred. It should be noted that neural networks were the preferred classifier for numeric input-based research. Although neural networks are claimed to achieve better results than other methods, they require both genuine and imposter samples during the training time, which may be impractical at the initial enrolment stage of the systems [16]. In recent years, support vector machines have become popular among researchers because of a perceived increase

in accuracy rate. Several authors have claimed that SVMs have competitive performance and less computational complexity than neural networks. However, Lee et al. [17] stated that SVMs perform poorly when the feature set is too large. Different experiments show that statistical models perform better for authentication (verification) tasks where machine-learning approaches perform better for user identification.

Different hybrid discriminative–generative approaches have been used in biometric systems, including speech recognition, object recognition, gait, human activity recognition, and handwritten text recognition. However, very few hybrid or combined approaches have been introduced so far in the area of keystroke biometrics. Table 3 shows the comparison of different existing keystroke biometric studies using hybrid/mix approaches.

**Table 3.** Comparison of existing hybrid approach-based research in KB systems.

| Reference Model | Users | Samples | Results |
| --- | --- | --- | --- |
| GA-SVM [45] | 21 | 3150–8400 | FAR: 0.00, FRR: 3.54 |
| GA-SVM [30] | 24 | 4200 | FAR: 0.43, FRR: 4.75 |
| PSO-SVM [30] | 24 | 4200 | FAR: 0.41, FRR: 2.07 |
| GA-BPNN [46] | 27 | 2700 | Accuracy: 88.90 |
| PSO-BPNN [46] | 27 | 2700 | Accuracy: 86.60 |
| ACO-BPNN [46] | 27 | 2700 | Accuracy: 92.80 |
| GMM-UBM [43] | 51 | 20,400 | EER: 5.50 |
| DNN [43] | 51 | 20,400 | EER: 3.50 |
| CPANN-SVM-DT [47] | 64 | 32,000 | Accuracy: 89.70 |
| CNN-SVM [48] | - | - | Accuracy: 92.1 |

FAR: False Acceptance Rate, FRR: False Rejection Rate, EER: Equal Error Rate.

Yu and Cho [45] proposed a GA-SVM wrapper approach for keystroke dynamic identity verification. A genetic algorithm was used for feature-subset selection and support vector machines for novelty detection. The model automatically selects a relevant feature subset and ignores the outliers. The best result was an average 3.54% FRR with the proposed GA-SVM approach. Yu and Cho's [49] further investigation on KB system research found that their previous experiments in KB systems had some limitations, such as lengthy training time for the model, the data preprocessing involving human interaction, and requiring a large data set. To mitigate these limitations, they proposed a hybrid approach using a genetic algorithm (GA) and SVM as a novelty detector and ensemble model-based feature selection to alleviate the deficiency of a small training data set. The experiment employed a GA paradigm for the randomized search and an SVM as a base learner in the wrapper approach. The overall idea of their proposed model is to use subsets of features that would evolve through the mechanism of the GA, to be evaluated through training and testing of an SVM with the data set. The authors claimed that the proposed hybrid approach has shown promising results in keystroke dynamics.

Azevedo et al. [30] presented the development of a hybrid system based on support-vector machines and stochastic optimization techniques. Genetic algorithm (GA) and particle swarm optimization (PSO) algorithms have been used for feature selection, and support vector machines have been used as verifiers. Using a GA evolutionary algorithm, the authors achieved a minimum error of 5.18% at a FAR of 0.43% and FRR of 4.75%. Using PSO, FAR and FRR were 0.41% and 2.07%, respectively, with a minimum error rate of 2.21%. The PSO evolutionary algorithm was suitable for the feature-selection task in their experiment. The experiment also suggested that PSO has a shorter processing time than GA. Karnan and Akila [46] proposed a personal authentication based on keystroke dynamics. They proposed ant colony optimization (ACO) for feature-subset selection and a backpropagation neural network (BPNN) for classification. The experiment compare the results with particle swarm optimization (PSO) and a genetic algorithm (GA). The experiment showed that ACO had better performance than PSO and GA in regard to features' reduction rates and classification accuracy.

Deng et al. [43] experiments in keystroke dynamics with a deep neural network (DNN) as a classifier performed better on a CMU data set. The research used negative samples for training the model, which might be a factor in achieving better accuracy. Deep neural networks are probabilistic generative models that consist of multiple hidden layers. The first step of DBN training involves layer-wise unsupervised pretraining of restricted Boltzmann machines (RBMs). One hidden layer can be trained at a time and the output of the lower-level layer serves as input to the higher-level layer. Finally, these pretrained two-layer generative models are collapsed into a single multilayer model, which serves as an initialized ANN for further discriminative parameter fine-tuning. DNN requires training with genuine and imposter data. The authors believe that the unsupervised generative training step in DNN gives the model good generalization capabilities for unexpected test data, and discriminative fine-tuning step gives better classification accuracy.

There are also some studies in KB areas that are combinations of various neural networks, statistical measures, and various pattern-recognition techniques. Dahalan et al. [50] suggested that combining fuzzy logic with a neural network may enhance the system's capability to learn the user's typing pattern. Teh et al. [51] proposed a fusion technique combining direction similarity measure and Gaussian probability density function, which enhanced the result with an EER of 9.96%. Raina et al. [52] proposed a hybrid approach in which a high-dimensional subset of the parameters is trained to maximize generative likelihood, and another small subset is discriminatively trained to maximize conditional likelihood. Their experiment found that the hybrid model provided a lower error rate and better accuracy than either purely generative or purely discriminative models.

## 3. Methodology

A literature survey on keystroke biometrics shows that discriminative models, especially artificial neural network (ANN) and support vector machine (SVM) classifiers, performed well and achieved better accuracy for large samples. Support vector machines have performed well for both user identification and verification [43]. Generative models have performed well with smaller data sets. The generative model POHMM [13] has performed better than the SVM and other models on the CMU benchmark short fixed-text data sets.

Besides pure generative and discriminative models, many researchers have also used hybrid or mixed approaches in biometric areas, especially in speech recognition [53], human activity recognition [54], sentence recognition, digit recognition, text recognition, cursive script recognition, signature recognition [55], and time-series prediction [56]. Lester et al. [57] showed that the combination of discriminative and generative classifiers (HMM with AdaBoost) was more effective than either of the classifiers on their own in modeling human activities. Fco. Javier et al. [53] compared the hybrid approach of ANN/SVM with HMM, and also found that the hybrid model achieved significantly better human activity-recognition performance. Rynkiewicz et al. [56] applied a hybrid HMM/ANN scheme to predict time-series data, which gave much better segmentation of the series. The following section discusses some existing hybrid/mixed approaches used in KB systems.

### 3.1. Proposed Model

Several studies [30,44–50] suggest that generative models outperform discriminative models for small data sets and can handle missing or irregular data. On the other hand, discriminative models have better asymptotic performance and perform well for large data sets and continuous authentication. A generative model, the partially observable hidden Markov model (POHMM) is a suitable tool to extract a user's typing features and is capable of handling not only stochastic processes but can also handle missing or irregular data. On the other hand, the discriminative model of the support vector machine (SVM) has performed well in various biometric areas for large data sets. In an attempt to take advantage of both generative models and discriminative models, this research proposes a hybrid POHMM/SVM model.

The architecture of the proposed hybrid POHMM/SVM model is presented in Figure 1. The system begins with extracting features from the raw data. The generative model POHMM is used to fit with the selected features. The POHMM model estimates and extracts POHMM parameters. Marginal distribution properties of the POHMM handle missing or novel data during likelihood calculation, while parameter smoothing handles missing or infrequent data during parameter estimation. Finally, the extracted POHMM parameters are used to train the SVM, build models, and compare with the test model. The POHMM acts as an unsupervised feature extractor for the SVM.

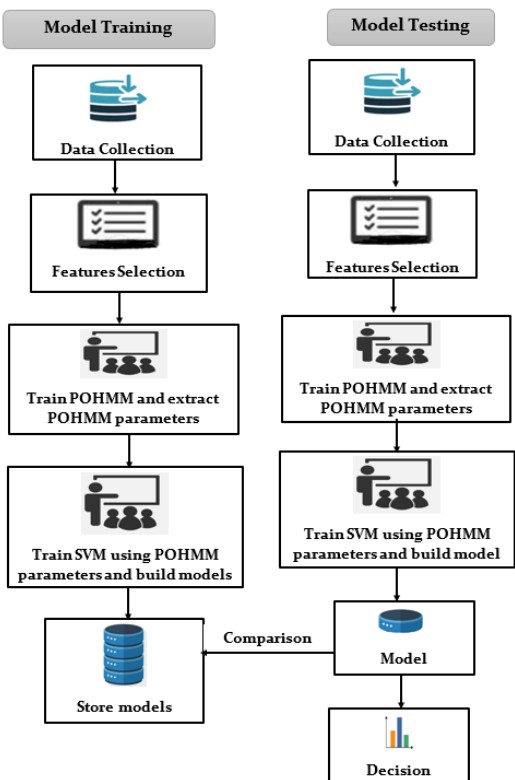

**Figure 1.** Proposed hybrid POHMM/SVM system overview.

The five main elements of the system are described below.

### 3.1.1. Raw Keystroke Data Collection

There are many experiments that have been conducted in keystroke biometrics. However, different studies have used different data sets and different features. They also differ in terms of evaluation procedures used. Some researchers have used benchmark data sets and some have used their own data sets. Therefore, it is hard to make comparisons of model performance with other work. To evaluate the proposed model, a CMU keystroke dynamics benchmark data set [14] has been used.

The CMU data set was chosen for evaluation due to its ability to provide a data set and analyze the performance of different existing keystroke dynamics algorithms for objective comparisons. The authors have evaluated the data set with fourteen existing keystroke dynamics classifiers, including Euclidean, Euclidean-normed, Manhattan-filtered, neural network-standard, fuzzy logic, and SVM-one class.

Keystrokes are detected by a keylogger that records and stores the sequence of keys typed by the users along with key-press and key-release timing information. Event times are measured in milliseconds with roughly 16-millisecond precision [58]. There were 51 subjects (typists) in the CMU keystroke benchmark data set, each typing a static strong password string: ".tie5Roanl". The data set also considers the Enter key to be a part of the password, making the 10-character password 11 keystrokes long. There were eight

data-collection sessions for each subject with at least one day between each two-session period. Fifty repetitions for the password string were collected in each session, resulting in 400 samples for each subject and a total of 20,400 samples for 51 users.

The CMU keystroke data set contains keystroke dynamics consisting of the dwell time (hold time) for each key, as well as the flight time between two successive keys—key press latency (KPL) and key interval (KI). Figure 2 shows the hold time of the keystrokes ".tie5Roanl" and Enter for the first two subjects and their first two sessions from CMU keystroke data sets. Each of the lines from each chart represents 50 samples per data-collection session. The features are highly correlated with large-scale variations and some are linearly dependent.

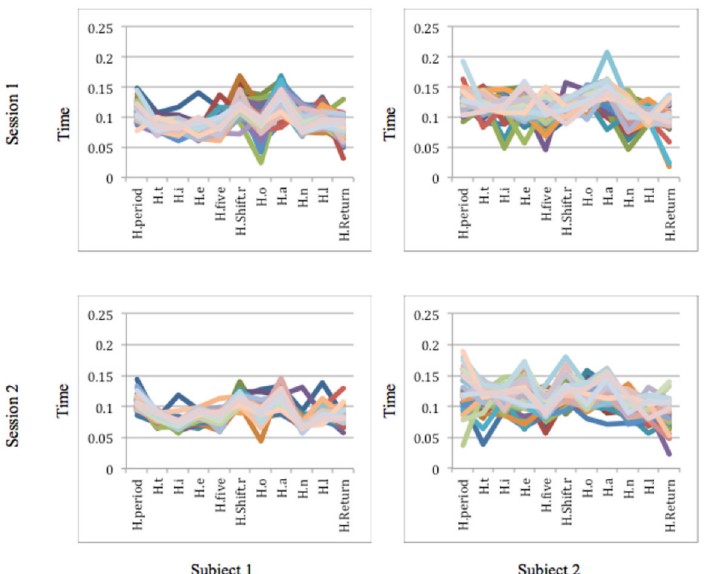

**Figure 2.** Hold time (HT) of 11 keystrokes in CMU keystroke benchmark data sets.

### 3.1.2. Feature Selection

In the two-state POHMM, a user can be either in an active state or a passive state of typing. The key names ".tie5Roanl Enter" are event types that partially reveal information about the hidden state. The timing feature vector for the CMU keystroke data set is formed by the 11-key hold (KH) time, 10 key-press latencies (KPL)/down-down, and 10 up-down/key interval (KI) latencies of the 11-keystroke sample, create a total of 31 timing features. Ten key-release latencies (KRL) and 10 release-press latency (RPL) features were also extracted using KH, KI, and KPL features. Therefore, a total of 51 dimensional feature vectors were extracted from the CMU keystroke benchmark data set.

For user identification and verification, the hybrid POHMM/SVM model uses hold time and key-press latency features. Hold time and key-press latency for each of the 11 characters are modeled by a lognormal distribution conditioned on the hidden state and the key name. Finally, they are multiplied by 1000 for normalization. Similar feature selection and normalization were used in [13]. The other features were extracted for comparison purposes in different experimental setups. Different studies have used different features or combinations of features. Hold time is the most used feature in keystroke biometrics. M.S. Obaidat [59] suggested that hold-time classification is superior to interkey time-based classification, and a combined hold-time and interkey time-based approach gave the least misclassification error. This research explores different features from the CMU keystroke data set and compares identification accuracy using different classifiers.

From two consecutive keystroke events, five types of features were extracted using the following formula:

$$KH_n = t_n^r - t_n^p \tag{1}$$

$$KI_n = t_n^p - t_n^r \tag{2}$$

$$KPL_n = t_n^p - t_{n-1}^p \tag{3}$$

$$KRL_n = t_n^r - t_{n-1}^r \tag{4}$$

$$KPL_n = t_n^r - t_{n-1}^p \tag{5}$$

where $KH_n$ denotes the key hold time, $KI_n$ is the key interval time, $KPL_n$ denotes the key press latency, $KRL_n$ is the key release latency, and $KPL_n$ is the release press latency of the $n$th keystroke. $t_n^r$ and $t_n^p$ are the release and press timestamps of the $n$th keystroke. Similarly $t_{n-1}^r$ and $t_{n-1}^p$ are the release and press timestamps of the $(n-1)$th keystroke.

### 3.1.3. Training POHMM and Extract POHMM Parameters

After feature extraction, the POHMM is trained and parameters are collected. The POHMM was developed and implemented by Monaco et al. [13], and the Python code can be downloaded from here [60]. The POHMM is an extension of the hidden Markov model (HMM), and the structure of the model is shown in Figure 3. $O_1^T = o_1, o_2, \ldots o_T$ represents the sequence of observation vector (emission vectors), $X_1^T = x_1, x_2, \ldots x_T$ is the sequence of observed value (event type), $\Theta_1^T = \theta_1, \theta_2, \ldots \theta_T$ is the sequence of hidden values (system state), and $T$ denotes the total number of observations. In the POHMM, the hidden state and emission depend on an observed independent Markov chain $X_1^T$. The emission $O_{t+1}$ depends on event type $X_{t+1}$ in addition to $\Theta_{t+1}$ and the hidden state $\Theta_{t+1}$ depends on $X_i$ and $X_{i+1}$ in addition to $\Theta_{t+1}$ [13]. For each sample, the POHMM is trained and parameters collected and stored in a file. The complete parameter estimation using a modified Baum–Welch reestimation algorithm, marginal distributions, and parameter smoothing is described as follows:

(a) Initialization: find initial parameters $\lambda$ and let $\overline{\lambda} \leftarrow \hat{\lambda}$
(b) Expectation: compute forward variable $\alpha_{j \mid Xn}(n)$, backward variable $\beta_{j \mid Xn}(t)$, posterior probabilities $\gamma_{jXn}(n)$ and $\xi_{ijXn,Xn+1}(n)$. Let $\overline{P} \leftarrow P(O \mid \lambda, X)$ where $O$ is the emission sequence, $\lambda$ is the model parameter and $X$ is the event type.
(c) Maximization: using the reestimation formula presented in [2,13], update the model parameters: initial state distribution $\pi$, state transition probability matrix $A$, and state emission probability matrix $B$, and let $\overline{\lambda} \leftarrow (\overline{\pi}, \overline{A}, \overline{B})$.
(d) Marginal distributions: find marginal distributions.
(e) Parameter smoothing: find smoothing weights and smooth the parameters with marginal distributions.
(f) Termination: if $P(O \mid \overline{\lambda}, X) - \overline{P} < \varepsilon$, then terminate and let $\overline{\overline{\lambda}} \leftarrow \overline{\lambda}$, otherwise go to step (b). $\varepsilon$ is the convergence criterion threshold.

For each training sample, POHMM provides a total of 130 parameters: 104 emission parameters and 26 transition parameters. POHMM provides 130 parameters for each training sample, and these parameters create a parameter vector for each sample that is used for identification and verification in hybrid models. For experimental purposes, three sets of parameter-emission parameters, transition parameters, and combined emission–transition parameters were collected.

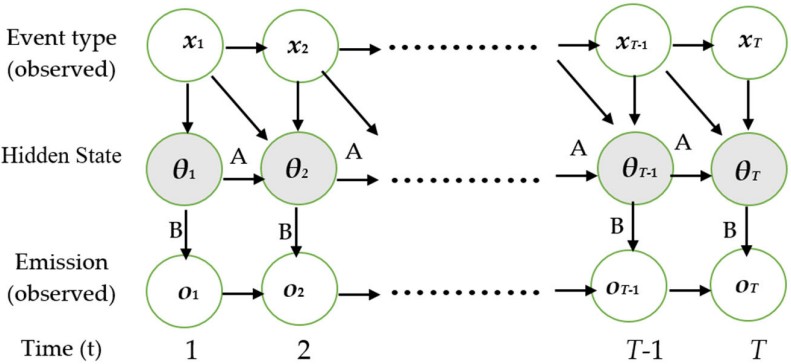

**Figure 3.** Structure of partially observable hidden Markov model, redrawn from [13].

### 3.1.4. Classifier Selection and Building Models

To address the classification problem, a support vector machine classifier was used in the proposed POHMM/SVM approach. The literature search revealed that the generative models performed better than the discriminative models for a smaller data set. However, for a larger data set, the discriminative models performed better than the generative models. To make the trade-off between small and large data sets, this research proposes a unique hybrid POHMM/SVM model. POHMM handles missing or irregular training data and SVM provides faster and better classification accuracy. POHMM was trained using keypress latency and key-hold time features for each sample from the CMU data set. After that, POHMM parameters were collected for each sample. These parameters were used for classification using the support vector machine, but for comparison purposes, we also examined POHMM parameters using other popular discriminative classifiers, such as *k*-NN, random forest, MLP NN, and logistic regression, in the experiment.

(a)    *Support Vector Machine*:

The support vector machine is a well-known supervised machine-learning algorithm generally used to solve two- or multiclass classification problems. However, for user verification, only the genuine data are available to train the model and a model has to be built for a genuine user only. Then, the model is used to detect an imposter user [50]. Schölkopf et al. [61] extended two- or multiclass SVM to one-class SVM to solve the one-class classification problem. The Scikit-learn 0.18.1 Python package and sklearn.svm module were used in the experiment [62]. We used the one-class SVM (svm.OneClassSVM) classifier in the proposed model for user verification. OneClassSVM is an unsupervised outlier detector that is based on the support vector machine library libsvm [63]. The components used for this classifier are radial basis function (RBF) as kernel function, 0.5 tolerance of training error, which means half the samples will become support vectors, and a kernel coefficient for "rbf" of 0.9 (gamma value). The CMU data set contains 51 unique users. For identification tasks, the multiclass SVM (svm.SVC) with "linear" kernel function was used. SVC is a C-support vector classifier based on the support vector machine library libsvm [64].

(b)    *k-Nearest Neighbor*

The *k*-nearest neighbor classifier is another frequently used classifier in biometrics. The *k*-nearest neighbor (*k*-NN) is a nonparametric classification method where assignment of a new class label to the input pattern is based on the nearest training samples in feature space. The *k*-NN is a simple classifier that requires only reference data points for both genuine and imposter classes. It uses data directly for classification without building a model first and does not requires any specific training phase. For a given unknown sample *f* and a distance measure, the nearest-neighbor rule for classifying *f* among *N* classes is presented below [64]:

1.  Find *k*-nearest neighbors from *M* training vectors without considering class label. Generally *k* in not multiple of *N* and chosen to be odd for a two-class problem.
2.  Find the number of samples $k_i$ from *k* neighbors and belonging to class $n_i$ where $i \in N$ and $\sum_{i=1}^{N} k_i = k$.
3.  The unknown sample *f* will be assigned to the class label $n_i$ with maximum $k_i$ number of samples.

The *k*-NN classifier was used for user identification using POHMM parameters. The Python implementation for *k*-NN classifier (KNeighborsClassifier) from the sklearn.neighbors module with all default parameters (number of neighbors = 5) was applied. The major drawbacks for any kind of *k*-NN based classifier are that the computing time is still longer than other classifiers and the performance is generally worse on high-dimensional data.

(c)  *Random forest*

Random forest [65] or random-decision forest is an ensemble learning method for classification. It is a class of ensemble method using decision-tree classifiers. Random forest is a mixture of tree predictors where every tree depends on the values of a random sample and with the equal distribution for all trees in the forest. Random forest has become popular in keystroke dynamics in recent years. The time required for training and testing using random forest is fast and achieves better accuracy in many applications. Random forest is an effective tool in prediction, but has been observed to overfit for some data sets with noisy classification. The RandomForestClassifier classifier and IsolationForest algorithm from the sklearn.ensemble module were used for identification and verification, respectively, in this experiment. We used default parameters for the RandomForestClassifier classifier, and the default number of trees in the forest was 10.

*Isolation Forest:* Isolation forest [66] outlier detection uses a random forest of decision trees for anomaly detection. Isolation forest or iForest builds an ensemble of iTrees for given samples and then samples with short average path lengths on the iTrees are considered as anomalies. The isolation forest algorithm isolates observations in two steps: (a) randomly select a feature, and then (b) randomly select a split value between the maximum and the minimum values of the selected feature. Isolation forest is ideal for large data sets because it has a linear time complexity with low constant and low memory requirement [66]. It also converges quickly with a small ensemble size, allowing high efficiency in anomaly detection. We used all default parameters for the iForest classifier (IsolationForest) from the sklearn.ensemble module.

*Logistic regression:* The logistic regression or the *logit* model [67] is a nonlinear transformation of the linear regression. The model is useful when dependent variables are limited to two-class problems [68] and generally calculates the class-membership probability for one of the two categories in the data set. The relationship between the predictor and the dependent variables in logistic regression can be written as:

$$\overline{p} = \frac{1}{1 + e^{-\overline{\mu}}}, \; where \; \overline{\mu} = \overline{\theta}.x \tag{6}$$

The prediction can be written in terms of $\overline{\mu}$, which is a linear function of *x*. $\overline{p}$ is used to predict genuine and imposter users. The decision boundary for logistic regression is also linear $x.\overline{p} = 0.5$. If the value of $\overline{p}$ is less than the threshold for a claimant user, then the user is genuine user; otherwise the user will be an imposter. Logistic regression is a popular classifier in the areas of medicine and bioinformatics. Logistic regression performs better than the decision trees and *k*-NN on continuous data sets [69]. This study used the logistic regression classifier LogisticRegression (aka logit, MaxEnt) from the Python sklearn.linear_model module for user identification.

(d)  *Multilayer perceptron neural networks (MLP NNs)*:

Multi-layer perceptron (MLP) neural networks are feed-forward ANNs used in pattern recognition, classification, and prediction. The backpropagation (BP) algorithm is the most

popular training technique used with MLP and has been applied in various fields, including network security, visual pattern recognition, handwriting recognition, medicine [70], intrusion detection, management, and finance. The performance of multilayer perceptron neural networks (MLP NNs) depends on various elements, such as number of input layers, number of neurons in each layer, the activation functions used by the neurons, and the choice of initial weights.

MLP is a supervised learning algorithm that learns a function by training on a data set. For a given set of features and a label, it can learn a nonlinear function estimator for classification. The difference between MLP and logistic regression is that there can be one or more hidden nonlinear layers between input and output layers. The most important advantages of MLP are that it is capable of learning a nonlinear model and can learn in real time. This experiment used the MLPClassifier class from [62] for identification. MLPClassifier implements a multilayer perceptron algorithm that trains with the backpropagation technique.

Besides the five discriminative models, the identification accuracy of the proposed POHMM/HMM model was also compared with the generative models HMM, POHMM, naïve Bayes, Gaussian mixture model (GMM), and Bayesian Gaussian mixture model (BGMM). To train/test HMM and POHMM, the procedures described in [13] were followed. For the naïve Bayes classifier, the GaussianNB class with default parameters was used from scikit-learn. For GMM and BGMM, the GaussianMixture class and BayesianGaussianMixture class were used, respectively, with a maximum of 100 iterations.

### 3.1.5. Model Training and Testing

A keystroke biometric system's performance is evaluated by how correctly the model can differentiate real users from attackers. The classification accuracy (ACC) is measured for model testing and evaluation purposes. Identification of the POHMM/SVM, POHMM/$k$-NN, POHMM/random forest, POHMM/MLP NN, and POHMM/logistic regression was performed as follows: key-hold time and key-press latency (KPL)/down-down features were extracted from the CMU data set. The POHMM model was trained with the 21 dimensional feature vector for each sample, and POHMM parameters were collected. There are two types of parameters extracted from POHMM: emission parameters and transition parameters.

The parameters are then split for training and testing data sets. Stratified $k$-fold cross-validation (SCV) is used to split data that randomly selects training and testing data sets. Stratified $k$-fold cross-validation provides training/testing indices to split data into training/testing sets. In $k$-fold cross-validation, the data set is partitioned into $k$ equal subsets. Each datum of the $k$ subset is used for the testing set, and the remaining $(k - 1)$ subsets are used for the training set. The cross-validation object StratifiedKFold is the variation of KFold that returns stratified folds, i.e., the fold preserves the percentage of samples for each class. The accuracy of each fold is determined and average accuracy of $k$-fold determined for overall accuracy. An example of stratified fourfold cross-validation is shown in Figure 4.

By training multiclass linear SVM with the training parameters, a system of $N$ models is created. For an unknown testing sample, the highest-likelihood class label is predicted. Finally, the accuracy score is determined for testing labels and data. The same evaluation procedure was followed for POHMM/$k$-NN, POHMM/random forest, POHMM/MLP NN, and POHMM/logistic regression, where the POHMM's parameters were used as features, and $k$-NN, random forest, and logistic regression were used as classifiers. We also evaluated the identification performance of $k$-NN, random forest, logistic regression, SVM (kernel = linear), SVM (kernel = RBF), MLP NN, and naïve Bayes on the CMU keystroke data set. Instead of POHMM's parameters, dwell time and flight time were used as features to train and test the models.

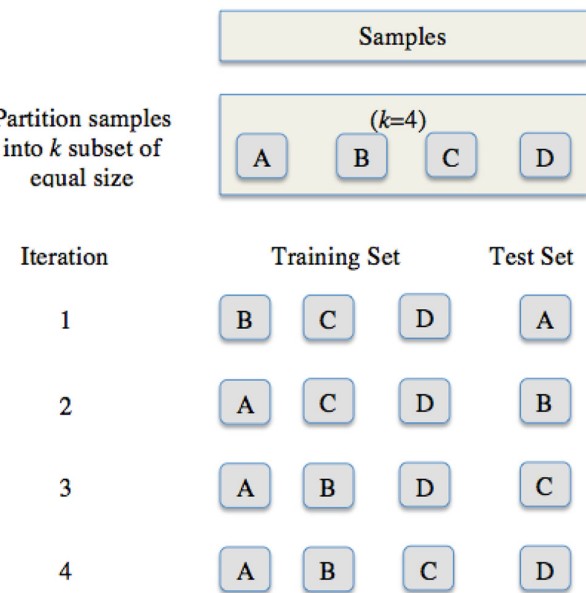

**Figure 4.** Example of stratified fourfold cross-validation.

## 4. Experimental Results

Experimental results were obtained in this study for the CMU keystroke fixed-text data set using the evaluation criteria presented in Section 3.1. To train POHMM, first a 2-hidden states POHMM model was generated using lognormal emissions and the frequency smoothing technique described in [71]. Lognormal distributions are used for time interval features, and other features are modeled by normal distribution. In the 2-state model, the user can be in either an active state or passive state. The key names have been used as passive state (p-state). The same procedure was followed for HMM, except there was no event type (key name). Identification performance of the proposed POHMM/SVM model was obtained, as presented in Section 3.1.5. A benchmark experiment compared the identification accuracy (ACC) of POHMM/SVM with the number of generative and discriminative models. In addition to the proposed hybrid POHMM/SVM, identification results were obtained from five benchmark anomaly detectors: naïve Bayes, k-nearest neighbors, random forest, multiclass SVM with linear kernel and RBF kernel, and logistic regression. Identification results were also obtained from hybrid POHMM/$k$-NN, POHMM/random forest, and POHMM/logistic regression models.

Key-press latency/down-down time and key-hold time features were used to train and test the models. The only difference in training/testing of hybrid models with purely generative/discriminative models is that purely generative/discriminative models are trained/tested with KPL and KH features, whereas in hybrid models, POHMM is trained with KPL and KH features first. The POHMM parameters are then used to train/test the classifiers. Identification results were obtained in this experiment using the stratified $k$-fold cross-validation described in the previous section with number of splits = 10 for all classifiers. Table 4 shows the identification results obtained from different generative, discriminative and hybrid models using key-press latency and hold-time features extracted from the CMU keystroke data set.

**Table 4.** Identification-accuracy comparison of proposed model with other models.

| Model | Accuracy, Mean (SD) | Max Accuracy |
|---|---|---|
| POHMM/SVM (kernel = linear) | 0.868 (+/−0.132) | 0.913 |
| POHMM/Logistic Regression | 0.827 (+/−0.135) | 0.876 |
| POHMM/MLP neural network | 0.825 (+/−0.144) | 0.879 |
| SVM (kernel = linear) | 0.809 (+/−0.106) | 0.865 |
| POHMM/*k*-NN | 0.789 (+/−0.151) | 0.854 |
| Logistic Regression | 0.777 (+/−0.152) | 0.842 |
| POHMM/SVM (kernel = RBF) | 0.762 (+/−0.180) | 0.822 |
| POHMM/Random Forest | 0.756 (+/−0.160) | 0.813 |
| MLP neural network | 0.764 (+/−0.163) | 0.849 |
| POHMM [13] | 0.748 (+/−0.151) | - |
| *k*-NN | 0.722 (+/−0.215) | 0.819 |
| Naïve Bayes | 0.696 (+/−0.122) | 0.777 |
| HMM [13] | 0.467 (+/−0.295) | - |
| SVM (one-class) [13] | 0.465 (+/−0.293) | - |
| SVM (kernel = RBF) | 0.216 (+/−0.134) | 0.308 |
| Bayesian Gaussian Mixture Model | 0.029 (+/−0.024) | 0.047 |
| Gaussian Mixture Model | 0.018 (+/−0.121) | 0.037 |

Features: key-press latency (KPL) and dwell time (HT). Stratified *k*-Fold (n_split = 10). Using combination of POHMM transition and emission parameters.

The identification comparison in Table 4 shows that POHMM/SVM (kernel = linear) outperforms the other models. The proposed POHMM/SVM model achieved the highest identification accuracy of 91.3% with average accuracy of 86.8% (standard deviation: 0.132) using both transition and emission parameters of POHMM. Tables 5 and 6 show the identification-accuracy comparison of different hybrid models on emission parameters and transition parameters of POHMM, respectively. This study also compared identification accuracy of different generative and discriminative models on the CMU short fixed-text data set using different features. Table 7 shows identification-accuracy comparison of different classifiers on different features extracted from the CMU benchmark keystroke short fixed-text data set. Accuracy of different models was achieved using the same stratified *k*-fold cross-validation with number of splits = 10. Standard deviations of the accuracy are estimated from the performance of each fold.

**Table 5.** Different hybrid models' accuracy on POHMM emission parameters.

| Model | Accuracy, Mean (SD) | Min Accuracy | Max Accuracy |
|---|---|---|---|
| POHMM/SVM (kernel = linear) | 0.862 (+/−0.130) | 0.685 | 0.913 |
| POHMM/Logistic Regression | 0.803 (+/−0.155) | 0.597 | 0.867 |
| POHMM/*k*-NN | 0.797 (+/−0.156) | 0.587 | 0.870 |
| POHMM/SVM (kernel = RBF) | 0.760 (+/−0.181) | 0.523 | 0.828 |
| POHMM/MLP neural network | 0.770 (+/−0.164) | 0.558 | 0.856 |
| POHMM/Random Forest | 0.750 (+/−0.161) | 0.597 | 0.867 |

Features: key-press latency (KPL) and dwell time (HT). Stratified *k*-Fold (n_split = 10).

Table 4 shows that all the hybrid models except the random forest achieved better accuracy than either of the purely generative or purely discriminative equivalents. Table 5 shows that POHMM/SVM (kernel = linear) achieved the highest accuracy of 91.3% (mean 86.2%) on POHMM emission parameters compared to other hybrid models. On the other hand, Table 6 shows that POHMM/SVM with linear kernel also achieved the highest identification accuracy of 74.5% (mean 68.9%) with the POHMM transition parameters than other hybrid models.

Table 7 shows that the linear support vector machine with linear kernel achieved better accuracy rate only for hold-time features. The random forest classifier achieved better identification accuracy for all other features or combinations of different features. For individual features, most of the classifiers achieved better accuracy for the key-hold

(KH) time feature. Again, combinations of different features provide better accuracy for all classifiers. A majority of the discriminative models achieved better identification-accuracy rates than the generative models.

**Table 6.** Different hybrid models' accuracy on POHMM transition parameters.

| Model | Accuracy, Mean (SD) | Min Accuracy | Max Accuracy |
|---|---|---|---|
| POHMM/SVM (kernel = linear) | 0.689 (+/−0.152) | 0.503 | 0.745 |
| POHMM/Logistic Regression | 0.616 (+/−0.149) | 0.430 | 0.681 |
| POHMM/*k*-NN | 0.657 (+/−0.156) | 0.461 | 0.731 |
| POHMM/SVM (kernel = RBF) | 0.612 (+/−0.168) | 0.411 | 0.689 |
| POHMM/MLP neural network | 0.653 (+/−0.155) | 0.458 | 0.718 |
| POHMM/Random Forest | 0.610 (+/−0.146) | 0.428 | 0.671 |

Features: key-press latency (KPL) and dwell time (HT). Stratified *k*-Fold (n_split = 10).

**Table 7.** Identification accuracy of different models using different features.

| Model | Accuracy, Mean (SD) | | | | | |
|---|---|---|---|---|---|---|
| | **Features: *HT*** | **Features: *KPL*** | **Features: *KI*** | **Features: *KPL, HT*** | **Features: *KPL, HT, KI*** | **Features: *KPL, HT, KI, KRL, RPL*** |
| SVM (kernel = linear) | **0.694 (+/−0.108)** | 0.548 (+/−0.154) | 0.603 (+/−0.174) | 0.810 (+/−0.106) | 0.811 (+/−0.106) | 0.811 (+/−0.104) |
| *k*-NN | 0.682 (+/−0.118) | 0.581 (+/−0.192) | 0.640 (+/−0.204) | 0.722 (+/−0.215) | 0.722 (+/−0.221) | 0.718 (+/−0.222) |
| Naïve Bayes | 0.668 (+/−0.120) | 0.268 (+/−0.087) | 0.325 (+/−0.081) | 0.696 (+/−0.122) | 0.646 (+/−0.121) | 0.578 (+/−0.101) |
| MLP NN | 0.655 (+/−0.141) | 0.445 (+/−0.153) | 0.604 (+/−0.187) | 0.764 (+/−0.163) | 0.779 (+/−0.147) | 0.750 (+/−0.158) |
| Random Forest | 0.648 (+/−0.104) | **0.646 (+/−0.163)** | **0.709 (+/−0.171)** | **0.827 (+/−0.135)** | **0.843 (+/−0.145)** | **0.841 (+/−0.151)** |
| Logistic Regression | 0.645 (+/−0.135) | 0.458 (+/−0.130) | 0.520 (+/−0.140) | 0.777 (+/−0.152) | 0.775 (+/−0.150) | 0.768 (+/−0.150) |
| SVM (kernel = RBF) | 0.033 (+/−0.009) | 0.101 (+/−0.074) | 0.066 (+/−0.051) | 0.216 (+/−0.134) | 0.207 (+/−0.146) | 0.257 (+/−0.176) |
| Gaussian Mixture Model | 0.020 (+/−0.027) | 0.017 (+/−0.011) | 0.022 (+/−0.019) | 0.018 (+/−0.021) | 0.017 (+/−0.022) | 0.023 (+/−0.016) |
| Bayesian Gaussian Mixture Model | 0.013 (+/−0.018) | 0.031 (+/−0.015) | 0.012 (+/−0.011) | 0.029 (+/−0.024) | 0.009 (+/−0.017) | 0.017 (+/−0.021) |

Stratified *k*-Fold (n_split = 10). Highest accuracy shown in bold. Using POHMM transition parameters.

## 5. Discussion

Keystroke biometric systems are still immature and less popular than other biometric systems. Recently, keystroke biometric systems have become an interesting research area, because they are cost-effective and relatively easy to integrate with existing security systems without adding any extra hardware. Researchers have proposed several KB systems, but they suffer from some shortcomings. Most of the studies have been conducted in a laboratory setting with subjects who had either expert or moderate typing speeds [72,73]. There have been no studies to date that quantify the KB system's performance for users with low typing proficiency or aged users. The keystroke biometric system has a strong psychological basis; therefore, a deeper understanding of typing behavior of people from different ages, genders, and backgrounds may enhance the accuracy and usability of the KB systems.

Many studies have collected data in fewer sessions with a short break in each session. This raises the question of whether a user's typing behavior changes with time. KB systems

are subject to template aging, as the typing behavior of users may change due to age, health conditions, and long-term behavioral changes. Most of the studies have utilized the same device for enrollment and testing. In some studies, the justification for a particular classification method remains low, with very little discussion on negative results. There are limited numbers of benchmark data sets for KB systems, and they have several limitations. One important limitation is that fewer users were involved in the data collection. This reduces the statistical impact of the results and makes it difficult to establish clear differences between the algorithms and the methods. Most of the data sets used the same password for all users, which is impractical for real-world application [74]. Finally, different research has used different performance measures, which makes it difficult to compare the performance of one model with another. There is a strong need to develop standardized protocols for evaluating and comparing KB system performance.

HMMs have shown greater success in biometric areas, especially in speech recognition. However, their performance is poor in keystroke biometrics compared with the other models. The POHMM, an extension of HMM, can capture better the underlying structure of data than the HMM, because the underlying system state that remains hidden is partially observable through the types of event in POHMM. For example, each individual key name is the partially observed events in short fixed text-based KB systems. Its marginal distribution property handles missing or novel data during likelihood calculation, and the parameter-smoothing technique handles missing or infrequent data during parameter estimation. On the other hand, SVM has been a popular model in several biometric areas for better classification performance. The proposed POHMM/SVM model inherits the benefits of both POHMM and SVM models and has shown promising performance in user identification.

A numbers of generative models, discriminative models, and hybrid models have been examined in this study for user identification purposes. The hybrid models POHMM/SVM, POHMM/random forest, POHMM/MLP NN, PPOHMM/$k$-NN, and POHMM/logistic regression were implemented and compared with different generative and discriminative models for identifying users. Table 4 clearly shows that the proposed hybrid POHMM/SVM model outperforms the purely generative and purely discriminative equivalents for user recognition. The hybrid POHMM/SVM (kernel = linear) approach outperformed over all generative and discriminative models, as discussed in the previous section, for identifying users. The experimental results show that by using both POHMM transition and emission parameters, hybrid models may achieve better identification accuracy than the purely generative or purely discriminative models. The discriminative linear SVM model achieved better accuracy for all different combinations of features than the SVM model with kernel RBF. However, the running time for linear time SVM is much longer than the RBF SVM.

The identification-accuracy comparison tables discussed above suggest a few important considerations:

(a)  In most cases, hybrid models achieved better accuracy than either their purely generative or purely discriminative equivalents.
(b)  POHMM emission parameters provided better accuracy than transition parameters. Using both transition and emission parameters produced better identification accuracy.
(c)  Discriminative models performed better than the generative models in user identification in the CMU data set.

## 6. Conclusions and Future Work

This research was inspired by other biometrics, such as speech recognition, signature recognition, and human activity recognition, where hybrid approaches have performed well. The main contribution of this research is developing a model for keystroke biometrics that will provide higher accuracy rates in accepting genuine users and rejecting imposters. The proposed hybrid POHMM/SVM model achieved improved identification accuracy compared to the other models for a short fixed-text data set. The proposed POHMM/SVM model inherits the benefits of both POHMM and SVM models, and that is why it has shown

improved identification performance. Besides KB systems, the POHMM/SVM model can also be used in other biometric applications, such as speech recognition, signature identification, modeling human activities, face and gesture recognition, and network anomaly detection. This model is also projected to achieve better performance in free-text data set, for continuous authentication.

One of the extended plans for this line of research will be to apply the proposed model with free text-based large data sets for user identification and verification purposes. This study used the same marginal distribution and parameter-smoothing technique used by Monaco and Tappert [13]. There is also room for future research to apply and compare with other parameter-smoothing techniques. A majority of the hybrid models in this study achieved better identification accuracy than the purely generative or purely discriminative models with the CMU short fixed-text data set. It will be interesting to examine the hybrid models, different generative and discriminative models with long fixed-text, long free-text, keypad, and mobile data sets. Future work might also examine the modification in the proposed hybrid model. POHMM can be used to model the temporal characteristics of the sequential data and the static classifier SVM that generates a posterior probability for each label.

There are many hybrid or mixed models that have been used in various biometric areas, including keystroke biometrics. However, to the extent of our knowledge, this is the first study in keystroke biometric where the generative model POHMM has been used as feature extractor and discriminative model SVM as classifier. Most of the existing hybrid models in KB systems are based on SVM and stochastic optimization algorithms, such as genetic algorithms and particle swarm optimization. Some studies have used a combination of distance measure classifiers, and some have used combinations of neurofuzzy algorithms, such as Fuzzy ARTMAP. The proposed POHMM/SVM approach uses a combination of pattern recognition and machine-learning-based algorithms for user identification and has shown promising performance. With increased privacy and security concerns, the proposed model can still be improved considerably, and can be used in the areas of cybersecurity and remote monitoring.

**Author Contributions:** Conceptualization, M.L.A., K.T. and M.A.O.; methodology, M.L.A., K.T. and M.A.O.; software, M.L.A.; validation, M.L.A., K.T., and M.A.O.; formal analysis, M.L.A., K.T. and M.A.O.; investigation M.L.A., K.T. and M.A.O.; data curation, M.L.A.; writing—original draft preparation, M.L.A., K.T. and M.A.O.; writing—review and editing, M.L.A., K.T. and M.A.O.; visualization, M.L.A., K.T. and M.A.O. All authors have read and agreed to the published version of the manuscript.

**Funding:** This research received no external funding.

**Conflicts of Interest:** The authors declare no conflict of interest.

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
