# Peer review of "A Hybrid Method for Keystroke Biometric User Identification"

_electronics, doi:10.3390/electronics11172782_

Round 1
Reviewer 1 Report
Greetings,
Authors have proposed
I view this paper as an application paper, where the author proposal handling the data, and applying several machine learning techniques in classifying Hybrid Method for Keystroke Biometric User Identification. The main contribution will be the framework itself but the technical contribution of this paper is slightly weak.
Do you apply or propose? Are there any changes being made, and why the combinations of them are adopted in this framework? Do you conduct any ablation study on this?
Table 1 and Table 2, is too unspecific and general. It should be more specific. Authors should clearly claim the particular contributions of their work respect to the references found in the state of the art, i.e. the contributions and limitations of those respect to authors' proposal are not clear.
Authors claim that " Most of the research have followed static authentication mode, which attempts to verify users at the initial interaction with the system. Only a few research were involved in dynamic authentication where the system will continuously or periodically monitors the keystroke behavior."
List the reasons for choosing the hybrid method for keystroke biometric user identification proposed by the authors.
3.1. Existing hybrid approaches in keystroke biometric areas, should be moved to literature review and discuss the implications of this approach.
Author mentioned on the parameters which can be used for POHMM and SVM , but there is no discussion/ analysis on why these parameters are used, so it appeared like the work is based on trial-and-error approach. Author may want to strengthen the contributions here.
Authors they have mentioned about POHMM, with respect to The k-Nearest Neighbors: and Random forest, and many more classifiers are used why they never mentioned in abbstract as well. Also, the performed stratified sampling process should be briefly detailed.
.Actually, the title of this section should reflect not only Feature selection but also the training/testing methodology.
The details about the methodology of application of the selected ML algorithms and their hyperparameters are missing. They should be included and argued. Do you conduct any ablation study on this? Are you manually perform the feature selection? A citation is needed in this part too.
Why not included specificity? The selection of one or another performance measure should be justified in terms of the Keystroke Biometric User Identification convenience.
Author Response
Hello,
Thank you for your valuable comments. Please see attached.
Kind regards,

Reviewer 2 Report
Keystroke biometric authentication is a supplement to conventional authentication methods. In the paper, the authors propose a hybrid ML method for keystroke authentication.
The paper has some problems. Keystroke biometrics can be used for 1) continuous authentication, in which the user has initially been authenticated by means of for instance password, and 2) keystroke biometric analysis on the typing of passwords. The authors make no clear distinction between these two categories to begin with. As far as page 8, line 292, it becomes clear that the authors are considering case 2.
There is a concern about novelty and contribution, which basically seems to be the chosen combined ML methods, POHMM and SVM. There are a number of ML methods, and the choice for POHMM/SVM is not clear. It appears that this was for some reason simply predetermined, and a rationale for this choice is therefore lacking. For a normal approach, the order would be to test a set of ML methods, and the chosen methods would be based on best performance. I am also curious about the individual performance of POHMM. According to Table 4, POHMM/SVM = 0.868 and SVM = 0.809 doesn't seem very significant.
Most of Section 5 Discussion does not discuss the proposed method and the performance result, lines 561-595.
Other issues:
* Sections 3.1 and 3.2.4 appears to be related work/background.
* The formulas on lines 329-333 look ugly. The English word must not be included in formulas. The formulas should be aligned by the equal-signs or left-aligned.
* The lambda symbol and the p symbol with the overlong bar on top (e.g. line 344) looks weird and must be replace by a proper-length bar.
* As a non-expert on machine learning, distinctions between terms such as generative / discriminative models and emission / transition parameters should be better explained.
* the continuing sentence after a formula (line 334) is not a new paragraph, and hence no indentation.
* There are numerous incomprehensible sentences. Here are a few:
- line 58: There are numbers of models have used to implement biometric application.
- line 144: Most of the research conducted in keystroke biometrics focused on verification compared to identification.
- line 272: Next, SVM is used to train the POHMM parameters as features, build models and finally compare with the test model.
* section x -> Section x
* line 348: 2.6.3 ??
* line 52: hard-to-hack passwords ??? Are some harder to hach than other? Do you mean hard
Author Response

(The authors gave the same response as above.)

Round 2
Reviewer 1 Report
Greetings,
The authors haven't colored the changes the reviewers asked for. So upload it.
Author Response
Hello,
Please see attached.
Thanks,

Reviewer 2 Report
Contribution and reproducibility
I am still concerned about the contribution. Simply combining two ML methods yields little if any contribution. Moreover do I feel that too much of the paper is generic ML material. It is also not clear how the proposed work is different from existing work, in particular considering performance. This should be indicated in Table 3. Of course, it is questionable how performance could be compared at all if different datasets are used, as pointed out by the authors.
More details of the proposed method should be included. Due to the generic nature of the paper and the highlevel generic presentation of the proposed methodology in Section 3, I have concerns about reproducibility. In Section 4 are presented some performance results for accuracy, but due to the lack of details, I cannot see how it is possible to reproduce and confirm the claimed performance results. Thus, in the end the claimed performance is just that, just claims.
I am also confused about the proposed setting of user identification. Why user identification and not user authentication? Since user identification is based on recognizing typing patterns of typed passwords, this simply implies there exists user authentication mechanism in which the user provides a username along with the password, which further implies that the proposed method would be a secondary security mechanism. Therefore I cannot see why or how identification is relevant at all, and why not authentication instead. The authors need to explain a rationale for this. Of course, it would give sense if the setting is user recognition on free texts.
Other:
* The abstract should not be written in past tense ("features have been extracted..", "the study examined.." ), but present tense. Also passive language should be avoided (see previous examples).
* Lines 341 is not a new paragraph, he hence lowercase "where".
* There is a problem with the subscripts on the lines 341-342 but not the lines after that.
Author Response
Hello,
Please see attached.
Thanks,
